# β-Cyclodextrin-Encapsulated Rhodamine Derivatives Core–Shell Microspheres—Based Fluorescent Sensor for Au^3+^ and Template for Generating Microplates of Gold

**DOI:** 10.3390/mi14071443

**Published:** 2023-07-18

**Authors:** Maniyazagan Munisamy, Balamurugan Rathinam, Esakkimuthu Shanmugasundaram, Vigneshkumar Ganesan, Vimalasruthi Narayanan, Suganya Bharathi Balakrishnan, Selvam Kaliyamoorthy, Stalin Thambusamy

**Affiliations:** 1Department of Industrial Chemistry, School of Chemical Sciences, Alagappa University, Karaikudi 630003, India; manichemist@gmail.com (M.M.); asmuthu92@gmail.com (E.S.); gvkumar.chemist@gmail.com (V.G.); annamsruthi2014@gmail.com (V.N.); sbsuganyaa@gmail.com (S.B.B.); 2Department of Nanotechnology and Advanced Materials Engineering, Sejong University, Seoul 05006, Republic of Korea; 3Department of Chemical and Materials Engineering, National Yunlin University of Science and Technology, Yunlin 64002, Taiwan; balar@yuntech.edu.tw; 4Department of Chemistry for Materials, Graduate School of Engineering, Mie University, Tsu 514-8507, Japan

**Keywords:** fluorescent core–shell microspheres, β−cyclodextrin, rhodamine derivative, Au^3+^ detection, bioimaging

## Abstract

We have developed β-cyclodextrin-encapsulated rhodamine derivative core-shell microspheres (β-CD@RH) to improve their aqueous solubility and biocompatibility. The β-CD@RH core-shell microspheres exhibited bright and stable fluorescence with Au^3+^ ion in aqueous media. The development of triangular and hexagonal gold microplates within an aqueous solution by a simple, one-step, and green chemistry strategy is followed and prepared. Fluorescent imaging of Au^3+^ in living cells is also successfully demonstrated.

## 1. Introduction

Gold (Au) has been one of the precious metals to humans since ancient times because of its esthetic qualities. Au^3+^ ions have been found to have a lot of interesting therapeutic properties and have been utilized to treat rheumatoid and TB arthritis in very small amounts [1]. However, the Au cations, Au^3+^ especially, are reported to be toxic toward human and aquatic species at high concentrations [2,3]. The toxicity of soluble Au^3+^ is due to its propensity to bind selectively to enzymes and DNA [4,5]. Au^3+^ is accountable for enzyme protein and depletion denaturation against selective cellular targets and lysosomal dysfunction, which results in the damage of DNA and membrane subsequently [6]. Au^3+^ also enhances oxidative DNA damage by catalyzing the production of free radicals in a variety of chemical entities employed in biochemical and biological studies [7]. To comprehend the real-world application of the sensors, the detection of Au^3+^ ions in complex and physiologically and ecologically significant solvents, in this case, phosphate buffer solution, was performed [8]. Therefore, the development of sensing tools that may be used for the selective detection of Au^3+^ over other interfering metal ions with very high sensitivity is highly desirable. In this regard, the literature contains a greater number of reviews on the detection of Au^3+^ in both aqueous and organic media [9,10]. Sensing Au^3+^ ion in an aqueous media is beneficial to identify the contamination in biological systems.

Among the various fluorescent probes designed for the recognition of Au^3+^, the fluorescence mode is advantageous due to its operational ultra-sensitivity and simplicity [11]. Moreover, the disaggregation-based fluorescence probe for Au (III) was simple to construct and showed excellent sensitivity, selectivity, and quick response time. Au (III) in cells can also be found using a probe or a paper-based strip [12]. Additionally, the latter complex binds to serum albumin and causes the typical CD bands to show in the visible spectrum. It has been shown that coordination at the level of the surface histidines is responsible for adduct formation for both of these gold (III) complexes [13]. The in-depth investigation of gold drug metabolism in vivo is studied the quick conversion of Au^3+^ to Au+ by thiol proteins in organisms. Here, they are addressed by an in vitro and in vivo luminescent Au^3+^ probe (RA) based on the ruthenium (II) complex [14].

Cyclodextrins are cyclic oligosaccharides that have a hydrophilic surface area and a hydrophobic central cavity, favoring the formation of inclusion complexes with low-polarity organic molecules [15]. However, compared to the cyclodextrin family, β-cyclodextrin is known for its give better applications in the pharmaceutical field, owing to a simple drug complexation at low costs. The formation of cyclodextrin inclusion complexes with either water-insoluble or poorly soluble drug/organic molecules can increase overall water solubility and stability, enhance bioavailability and decrease the toxicity of related substances [16]. Moreover, recently most researchers have reported that microplates are used in different applications such as tissue engineering, biosensor, bioimaging, and pharmaceutical industries due to the controllable structures can enhance the hydrophilic and hydrophobic nature, high surface-to-volume ratio, and biocompatibility.

Many classes of molecular fluorescence chemo sensors have already been reported for metal ions detection [17,18]. Spiro lactam rhodamine-based chemo sensors have recently gained a lot of attention due to their great fluorescence characteristics, such as long absorption and emission wavelengths as a dye, good absorption coefficient, and high fluorescence quantum yield [19]. Research on the advancement of fluorescent probes for the selective detection of Au^3+^ has been gaining a lot of interest. Nevertheless, various flaws in these fluorescent sensing systems, such as cross-sensitivity to other metal cations, low water solubility, sluggish response, low fluorescence cytotoxicity, and enhancement, obstruct their optional functioning. There is a need to create new fluorescent probes for gold ions that may overcome these limitations.

In the present work, we have synthesized β-Cyclodextrin-Rhodamine derivative (β-CD@RH) based core-shell microspheres through a hydrothermal process (Figure 1). The β-CD@RH core-shell microspheres exhibited good stability in an aqueous phase. Moreover, the β-CD@RH core-shell microspheres showed high sensitivity and selectivity towards Au^3+^ in an aqueous medium as verified by fluorescence studies and therefore the present probe may serve as a fluorescent sensor for recognition of Au^3+^ in living cells. To identify the β-CD@RH core-shell microspheres, we redirected the microsphere as an interior template to cultivate Au triangle and hexagonal microplates.

## 2. Materials and Methods

Rhodamine derivative (RH) was synthesized based on the reported procedure [20]. Rhodamine derivative (RH) was obtained by refluxing 2-pyridinecarboxaldehyde with rhodamine B hydrazide, in 77% yield (Scheme S1, ESI†). Synthesized fluorescent probe RH was further confirmed by standard spectroscopic techniques such as 1H and 13C NMR spectroscopy, as well as ESI-MS (Appendix A, ESI†). Further, the framework of RH was as certified by X-ray diffraction analysis. RH crystallized in the P21/c space group, which is monoclinic (CCDC no: 1440773) (Appendix A, Appendix A, ESI†). An aqueous solution of β-CD (1 mmol) and an ethanol solution of RH (1 mmol) was mixed for 6 h at room temperature in a traditional experiment for the synthesis of β-CD@RH core-shell microspheres. After that, the β-CD@RH solution was transferred to a Teflon-lined stainless-steel autoclave (100 mL) and heated to 180 °C for 24 h. Scanning electron microscopy (SEM) and transmitting electron microscopy (TEM) were used to examine the generated β-CD@RH core-shell microspheres.

### 2.1. General Information and Materials

All of the materials for synthesis were purchased from commercial suppliers and used without further purification. The absorption spectra were recorded on a Shimadzu UV-PC-2401 UV-vis spectrophotometer using 10 mm path length quartz cuvettes in the range 300−800 nm wavelengths, while the fluorescence measurements were carried on a Horiba Jobin-Yvon FluoroMax-4 spectrofluorometer using 10 mm path length quartz cuvettes with a slit width of 5 nm at 298 K. FT–IR spectra were measured on a JASCO FTIR 4600 FT–IR spectrometer with 4 cm^−1^ resolution and 32 scanned between wave number 4000 cm^−1^ and 400 cm^−1^. Samples were prepared KBr disks with 1 mg of complex in 100 mg of KBr. The mass spectra of RH using Agilent Technologies 6520 Accurate mass spectrometer. NMR spectra were recorded on a Varian FT-400 MHz instrument. The chemical shifts were recorded in parts per million (ppm) on the scale. The following abbreviations are used to describe spin multiplicities in ^1^H NMR spectra: s = singlet; d = doublet; t = triplet; m = multiplet.

### 2.2. Synthesis of the Sensor Molecule (RH)

Rhodamine-B hydrazide (0.46 g, 1 mmol) was dissolved in 20 mL absolute ethanol. An excessive 2-Pyridinecarboxaldehyde (4 mmol) was added, and then the mixture was refluxed in an air bath for 6 h. After that, the solution was cooled (concentrated to 10 mL) and allowed to stand at room temperature overnight. The precipitate which appeared next day was filtered and washed 3 times with 10 mL of absolute ethanol. After drying under reduced pressure, the reaction afforded 0.47 g of RH (77%) as pink solid, mp (°C): 218 ± 2. ^1^H NMR [CDCl_3_), SiMe4, J (Hz), δ (ppm)]: ^1^H NMR (300 MHz, CDCl_3_) δ 8.46 (d, J = 5.8 Hz, 1H), 8.00 (d, J = 8.1 Hz, 2H), 7.60 (t, J = 7.7 Hz, 1H), 7.52–7.40 (m, 2H), 7.12 (t, J = 6.5 Hz, 2H), 6.55 (d, J = 8.8 Hz, 2H), 6.45 (d, J = 2.4 Hz, 2H), 6.24 (dd, J = 8.8, 2.5 Hz, 2H), 3.31 (q, J = 7.1 Hz, 8H), 1.14 (t, J = 7.1 Hz, 12H). ^13^C NMR (75 MHz, CDCl_3_) δ 165.61 (s), 153.43 (s), 152.60 (s), 145.97 (s), 136.30 (s), 128.46 (s), 108.58 (s), 105.85 (s), 98.54 (s), 77.66 (s), 77.23 (s), 44.52 (s), 12.61 (s). ESI mass: calculated value for [(C34H35N5O2) H] (M + H) is 545.67, experimental value 546.3 (Appendix A, ESI†). The displacement ellipsoids created at the 30% probability level clearly show the X-ray crystal structure of RH evident. And with the crystal packing of RH, the hydrogen atoms have been eliminated for clarity. For clarity, intermolecular interactions have an outing of that (Appendix A, ESI†).

### 2.3. Synthesis of β-CD@RH Microspheres

The β-CD@RH core-shell microspheres were prepared according to the following procedure. Typically, 1 mmol of β-Cyclodextrin was dissolved in deionized water (solution A). One mmol of RH dye was dissolved in 10 mL acetonitrile. The obtained solution was added into solution A drop by drop and stirred for 6 h under ambient conditions. The prepared clear solution was then transferred into a 100 mL Teflon-lined stainless-steel autoclave and maintained at 180 °C for 24 h. After cooling to room temperature, a brown suspension was obtained, which was centrifuged at 8000 rpm for 10 min and washed three times with DI water and ethanol. Finally, the precipitate was collected and dried in an oven at 80 °C overnight, which is noted as β-CD@RH microspheres.

### 2.4. Cell Culture

The Mesenchymal stem cells (C3H10T1/2) were purchased from National Centre for Cell Sciences (NCCS), Pune, India. The cells were plated at 7 × 105 per 25 cm^2^ flask, maintained in Dulbecco’s modified Eagle’s medium (DMEM) supplemented with 10% fetal bovine serum (FBS) and 1% Penicillin/Streptomycin (10,000 U/mL) and maintained in a humidified incubator at 37 °C and 5% CO_2_.

### 2.5. In Vitro Cytotoxicity

The cytotoxicity of the β-CD@RH microspheres, Au^3+^ and β-CD@RH-Au^3+^ were studied using MTT (3-(4,5-dimethylthiazol-2-yl)-2,5-diphenyltetrazolium bromide) assay on Mesenchymal stem cells. In brief, 10,000 cells/well in 100 µL final volume was seeded into 96 well plates, once the cells were 70% confluent the media was removed and washed with PBS (Phosphate buffer saline) and replaced with serum-free media. The sterilized films (*n* = 5) were placed gently into each well above the cells and wells without the β-CD@RH microspheres were regarded as control. After 24 h of incubation, the wells were emptied and 10 µL of MTT reagent from stock (5 mg/mL in PBS) was added to each well reaching a final concentration of 0.5 mg/mL. Then the cells were incubated for 2 h at 37 °C in a 5% CO_2_ Incubator. After incubation the wells were emptied again and 200 µL DMSO was added to each well to dissolve the insoluble purple formazan crystals. The 96 well plate was placed on a rocker for 20 min and then measured spectrophotometrically in an ELISA reader at a wavelength of 570 nm.

## 3. Results and Discussion

The SEM images of the β-CD@RH core-shell microspheres are shown in Figure 1a,b. The average diameter of the CD@RH core–-hell microspheres is 650 to 900 nm. An SEM picture of β-CD@RH microspheres shown in Figure 1b, 2 μm diameter is the enlarger image. It is evident that the microsphere SEM image. The β-CD@RH core-shell microspheres were dispersed perfectly in an aqueous solution; no aggregation was noticed due to the electrostatic repulsion power between the β-CD@RH core-shell microspheres. TEM was used to evaluate the morphology and size of the β-CD@RH core-shell microspheres and the TEM pictures are shown in Figure 1. As shown in Figure 1c, the TEM picture of β-CD@RH core-shell microspheres shows that the β-CD@RH microspheres are spherical with great distribution, and the average size of the β-CD@RH is 0.46 to 0.52 μm. Then, measured the particle size distribution of the material, which is observable at 346 nm, and measured the mean physical size of the particles. DLS measurements showed that the average of the β-CD@RH core-shell microspheres was 750 nm, and this particle size value is a bit bigger than our particle size distribution result. The histogram highlighted in yellow in Figure 1d is part of the manuscript and have included in Appendix A, ESI†. The Zeta potential evaluation can be an imperative factor for checking the balance of β-CD@RH core-shell microspheres within the aqueous medium. The findings of the zeta potential experiments for β-CD@RH core-shell microspheres are shown in Appendix A (ESI†). As is seen in Appendix A (ESI†), the zeta potential value for β-CD@RH core-shell microspheres were found to be −42.2 mV. The β-CD@RH core-shell microspheres possess a sizable negative zeta potential value, as the particles are likely to repel one another and have improved suspension stability thereby. It is obvious that the probe β-CD@RH core-shell microspheres were colorless in an aqueous solution, had no apparent absorption, and was non-fluorescent, indicating that the spirocyclic form of the dye was dominant. The colorless and non-fluorescent solution progressively became pink in color and began to fluorescent under UV light with the addition of Au^3+^ (50 mM, 1 equivalent, regarding probe β-CD@RH). With the addition of Au^3+^, a new absorption band located at 556 nm (Figure 2a) and a strong fluorescence band at 582 nm (Figure 2c) related to the delocalized xanthene moiety of rhodamines appeared. It is important not to rule out the feasible involvement of additional metal ions in the sensing process. Therefore, the titration of the probe β-CD@RH with various other metal ions was completed under the same sensing conditions. To our delight, related metal ions such as Ni^2+^, Hg^2+^, Co^2+^, Fe^2+^, Ca^2+^, Cd^2+^, Mg^2+^, Cu^2+^, Al^3+^, and Fe^3+^ did not trigger any ring-opening reaction to lead to a spectroscopic response. Only in the presence of Au^3+^ did the fluorescence and absorption spectra alter significantly. Moreover, no significant color or spectral changes in β-CD@RH solution were noticed with the addition of other metal ions demonstrating the high selectivity of the probe towards Au^3+^ ion. The RH is a dye that contains nitrogen; typically, nitrogen has a lone pair of electrons, making it most likely to impart a negative charge to the microspheres. Since metal ions have potential, the negative of RH is that ready attracted by a positive metal ion charge.

The titration of Au^3+^ against probe β-CD@RH core-shell microspheres gave a strong absorbance and fluorescence improvement with the addition of the Au^3+^ ions (Figure 2b). In the concentration range of 1–100 mM, the fluorescence titration profile of the probe β-CD@RH core-shell microspheres with Au^3+^ showed a strong linear relationship, showing that the probe β-CD@RH is suitable for quantitative evaluation. The detection limit was determined to be 0.6 parts per million. The enhancement of emission intensity was not observed above 100 mM of Au^3+^. Although the spectroscopic response of the β-CD@RH core-shell microspheres was quite sluggish, the entire saturation of the fluorescence intensity strength was observed after 60 min. The β-CD@RH solution showed intense fluorescence with an approximately 10-fold enhancement in the fluorescence intensity at 590 nm. That is the fact the β-CD@RH could use as an off-on fluorescent chemosensor for Au^3+^.

A pH investigation was carried out to regulate the performance of the fluorescent probe β-CD@RH by optimizing the pH of the experimental condition. In the absence of Au^3+^ ions, the fluorescent probe, β-CD@RH exhibited fragile fluorescence and showed pH independency over the pH range 4.0–10.0 (Appendix A, ESI†). At low pH, the probe demonstrated high emission intensity because at low pH the spirolactam band opens irrespective of metal ions added. Nevertheless, it is noticed that the presence of Au^3+^. However, it has here shown the emission intensity of β-CD@RH at pH 4.0–10.0 is selectively increased by the presence of Au^3+^ ions. By keeping the β-CD@RH core-shell microspheres and Au^3+^ ions constant (10 mM) and altering the mole fraction of Au^3+^ ions from 0 to 1, the Job’s plot method for the absorbance was used to calculate the binding stoichiometry of the β-CD@RH-Au^3+^ complex (Appendix A, ESI†). The absorbance maxima moved through a maximum at a molar fraction of nearly 0.5, suggesting that a 1:1 stoichiometry is most likely for the binding mechanism of β-CD@RH core-shell microspheres and Au^3+,^ according to the Job’s plot. The above results illustrate that the probe β-CD@RH core-shell microspheres exhibited an excellent binding capability towards the Au^3+^ ions. The optical properties of the materials using the absorption and emission range of β-CD@RH and β-CD@RH-Au^3+^. The molar extinction coefficient (ε max) and fluorescence quantum yield (Φ_F_) are displayed in Appendix A. The β-CD@RH-Au^3+^ has a high ε_max_ value compared to the β-CD@RH, and the β-CD@RH complex has a higher fluorescence behavior following the addition of Au^3+^, as indicated by the Φ_F_ value.

In order to ascertain the selectivity of the probe toward Au^3+^, several competitions experiments were performed where the probe β CD@RH core-shell microspheres were utilized for the sensing of Au^3+^ in the presence of other competing metal ions (Appendix A, ESI†). The results indicated that these ions showed no apparent interference in the Au^3+^ detection. Thus, the β-CD@RH core-shell microspheres exhibited excellent selectivity toward Au^3+^, which is essential for biological sensing applications. The entire sensing process is suggested to be reaction based and irreversible. To confirm this argument, a surplus quantity of S^2-^ ion (Na_2_S (100 mm)) was put through the colored and fluorescent solution containing Au^3+^ and probe β-CD@RH. There were no noticeable changes either in the absorbance or fluorescence intensity after the addition of the sulfide ion, which indicates that the sensing process is irreversible (Appendix A, ESI†). In particular, when a rhodamine derivative adjoined with other organic moieties gives rise to a rhodamine derivative that may be enduring in two isomeric forms; one of the isomers exists as spirolactam with open ring structure (extended large π-conjugation arrangement) another one being ring closed (limited π-conjugation arrangement). However, the optical properties of both isomers are entirely different in that the ring’s closed form is non-fluorescent and colorless. Furthermore, the association of metal ions transforms the specific spirolactam ring open which gives intense color and fluorescence through extended π-conjugation between the rhodamine derivative and metal anion. This notable characteristic furnishes an ideal model for conserving the OFF-ON mechanism. Here, β-cyclodextrin used as a carrier or core-shell of rhodamine derivatives, have a peculiar physicochemical property in its hydrophobic interior and hydrophilic exterior. Those properties and behaviors capable of sensing the Au+ ions may be due to the presence of outermost electronic configurations. The limit of detection (LOD) was determined in response to your suggestion using the following formula:LOD = 3.3 (Sy/S)(1)

Sy is the response’s standard deviation, and S is the slope of the linear curve. As a result, when the relationship between Au (III) concentration level and absorbance intensity, the graph plotted and obtained the linearity, and this curve to finding the LOD, this is shown in Appendix A. The β-CD@RH core-shell microspheres detection of Au (III) the LOD value is 38 nm. Recently, Atul Kapoor’s research team members have proposed the AIEE active Azomethine-based rhodamine derivative for fluorimetrically and electrochemically LOD is 40 nm [21]. Here we also report the β-CD has introduced detection limit decreased, and the solubility is enhanced.

When the β-CD@RH-Au^3+^ solution is slowly aged, an appreciable quantity of precipitate was produced at the bottom of the vial. As the focus exceeds the templating character of the β-CD@RH core-shell microspheres alter their work as an exterior template to create Au microplates. The assumption is that the nucleation of Au ions was initiated on the top of β-CD@RH core-shell microspheres when it was completely saturated with Au ions. Once Au nuclei were created on its surface, they grow by consuming the Au^3+^ ions around them continuously and giving the Au microplates. As shown in Figure 3a–d, SEM photographs of the sample revealed that the gold microplates are mostly triangular and hexagonal shapes. The average size of the microplates is 0.9–1.3 μm. The microplates retain their shape and are stable for several days. The β-CD@RH core-shell microspheres play an important role in the creation of Au microplates, according to this research.

Encouraged by all these attractive properties as evidenced by spectroscopic data that the probe can selectively sense Au^3+^ in an aqueous medium with a quick response, and also, evaluated the features of the probe β-CD@RH core-shell microspheres for imaging Au^3+^ in living cells. The typical MTT assay showed that the probe β-CD@RH microspheres at the micromolar concentrations exhibited no cytotoxicity to the cells, and therefore it could be utilized as a probe for cell imaging. Mesenchymal Stem Cells (MSCs) (C3H10T1/2) were treated with probe β-CD@RH core-shell microspheres for 30 min before adding Au^3+^ and incubating for another 30 min. The fluorescence pictures were taken before and following the addition of Au^3+^ (10 mM) (Figure 4). MSCs incubated with probe β-CD@RH core-shell microspheres and Au^3+^ showed no fluorescence, whereas cells stained with probe β-CD@RH-Au^3+^ showed a strong fluorescence signal, indicating that the probe β-CD@RH core-shell microspheres is cell membrane permeable and can be efficiently used for in vitro imaging of Au^3+^ ions in living cells. Furthermore, there were no such indications of cell harm by β-CD@RH core-shell microspheres. The cells were intact and showed healthful adherent and also passed on morphology after and during the labeling procedure with probe β-CD@RH, indicating a lack of cytotoxic effects (Appendix A, ESI†). To better understand various metabolic processes and increase disease detection, it is essential to quantify the concentration and track the distribution of metal ions in individual cells, organs, and entire organisms. Among the various techniques for metal ion detection, fluorescent sensors using β-cyclodextrin-encapsulated rhodamine derivatives have drawn a lot of interest because of their many advantages, including operating simplicity, high sensitivity, quick detection speed, and real-time detection. This research highlights recent advancements in metal ions in biological systems, especially Au^3+^.

## 4. Conclusions

In conclusion, we have developed a β-CD@RH core-shell microsphere based on rhodamine derivative and β-cyclodextrin describing its extremely selective and sensitive fluorescence sensing nature towards Au^3+^ ions in an aqueous medium for the first time. The fluorescent probe β-CD@RH core-shell microspheres can be used to sense selectively Au^3+^ ions over other biologically relevant metal ions. The β-CD@RH core-shell microspheres could serve as a template to create a triangle and hexagonal gold microplates. The successful demonstration of Au^3+^ fluorescence imaging in living cells has been performed in this study. We anticipate the β-CD@RH core-shell microspheres could make a fluorescence probe for tracking Au^3+^ ions. Last but not least, the NMR analyses validate the development of the fluorescent probe RH. The packing of RH’s crystals is also better understood because of the X-ray crystal structure and analysis. These studies have been by Dynamic Laser Scattering, Zeta Potential, competition experiments between Au^3+^ and specific metal ions, and the Cytotoxic effect.

## Data Availability

The data presented in this study are available on request from the corresponding author.

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
