# Peer review of "β-Cyclodextrin-Encapsulated Rhodamine Derivatives Core–Shell Microspheres—Based Fluorescent Sensor for Au3+ and Template for Generating Microplates of Gold"

_micromachines, 2023, doi:10.3390/mi14071443_

Round 1

Reviewer 1 Report (New Reviewer)

1. The Materials and Methods is too brief. The author should provide detailed, step-by-step protocol, for example, what is "in a traditional experiment for the synthesis of β-CD@RH core–shellmicrospheres"? What is the volume used for the complex formation (only concentration were provided)? What are the references that the authors followed for the core-shell synethsis (why using autoclave)? A good manusciprt should include good Methods part that other researchers can follow and reproduce the results.

2.The author should provide additional evidence to support that the beta-CD@RH micorspheres contains the rhodamine inside, it could be that the dye is still in the particle solution or it only attached to the particles' surface.

3. Additional control experiment should be performed to study the effect of Au3+ on the rhodamine molecule alone: what's the aborption & fluorence changes of RH with the addition of Au3+? It could be that the particle solution contains RH and interact with Au3+, not the microsphere.

4. Regarding the cell imaging: the microsphere has a size of >650 nm, and negative potential on its surface, these are all conditions that make the microsphere less favorable for cell permeablization.  The figure 4b&c can't not prove the microsphere is cell permeable, it could be that the Au3+ renders positive charge to the microsphere and helps it get into the cell, and therefore, it can not be claimed as a Au3+ detector used in living cells (as it requires the presence of Au3+ to get into cells). Additional experiment is required to prove that the microsphere is cell permeable.

The English has to be improved.

Author Response

Reviewer 2 Report (New Reviewer)

In this manuscript, the authors report the preparation and chemo-sensing function of b-CD@RH core-shell microspheres. They successfully demonstrated the highly selective fluorescence sensing of Au3+ using the microspheres. They applied the sensing function to cell imaging and also found the formation of gold microplates on the core-shell microsphere. Such easily prepared micromaterials with sensing functions can be potentially applicable in the various scientific and technological fields. In my opinion, the manuscript is acceptable for publication in Micromachines after the following points are addressed.  

1) How does the b-cyclodextrin in the core-shell microsphere contribute to the high Au3+ selectivity against other metal ions in the fluorescence sensing? The authors should describe the mechanism of the selective sensing in more detail.

2) The authors are recommended to briefly describe the potential application of the gold microplates generated on the core-shell microspheres.

3) Page 3, line 86 “As shown in Figure 1 (c) and (d)”

The image (d) is missing in Figure 1.

Spaces are missing in some parts of the text. Please carefully check the manuscript before submission.

Page 2: “ismonoclinic”

Page 3: “werecolourless” “(Figure. 2c)related” “sensingprocess” “sensingconditions”

Page 4: “thesulfide”

Page 5: “Auions” “vitroimaging”

etc. 

Author Response

Reviewer 3 Report (New Reviewer)

In this manuscript, the authors propose the use of β-Cyclodextrin-Encapsulated Rhodamine Core–Shell Microspheres as a sensor for Au3+ ions. While the topic of the manuscript aligns with the journal's scope, significant revisions are necessary before considering it for publication in Micromachines. The following points need to be addressed:

1.    There seems to be confusion between rhodamine and rhodamine derivatives in the manuscript. It is crucial to clarify whether the research is based on a rhodamine derivative (RH) rather than rhodamine B to avoid misleading the readers. Additionally, the title may potentially mislead readers and should be reviewed accordingly.

2.    Figure 1 presents SEM and TEM images, which are well-executed. However, further analysis is required to include the size distribution and the PDI value derived from SEM, TEM, and DLS measurements. Additionally, it should be noted that there is no Figure 1 (d), and line 86 needs to be revised accordingly.

3.    In Figure 2, the optical properties of the beta-CD@RH microspheres for sensing different metal ions are displayed. To provide a comprehensive understanding, titration curves depicting the relationship between Au(III) concentration and absorbance or fluorescence intensities should be included. Furthermore, it is important to clarify whether the range of 0 to 2 equiv. indicates the concentration of Au(III) ions relative to beta-CD@RH microspheres or the RH concentration.

4.    Figure 2 (d) claims that the fluorescence intensity increases with the concentration of Au(III) ions. The authors should provide an explanation as to why the fluorescence peak maximum shifts to shorter wavelengths (blueshifts) when the concentration of Au(III) ions increases.

5.    The manuscript suggests the possibility of utilizing beta-CD@RH microspheres as a probe for cell imaging in Figure 4. However, it is necessary to discuss how and where the beta-CD@RH microspheres label the cells and why Au(III) ion probing is required in this context. Figure 4 (c) depicts the incubation of MSC cells with beta-CD@RH microspheres in the presence of Au(III) ions, and this should be further elaborated upon.

These revisions are essential to improve the clarity and scientific rigor of the manuscript, which are necessary steps before considering it for publication in Micromachines.

Round 2

Reviewer 1 Report (New Reviewer)

I retains my opions that significant revisions are necessary before considering it for publication in Micromachines. 

The author didn't provide any control experiment request in my review, and just reply with "we all believe that the microsphere is cell permeable without the need for any additional experiments" is not acceptable.

The quality has been improved

Author Response

Please see the Cover letter

Reviewer 3 Report (New Reviewer)

Thank you for the improvements made to the manuscript. However, there are still some areas that require clarification.

Regarding Figure 1, further analysis is needed for the size distribution and the PDI value, which indicates the degree of size homogeneity. It is requested to measure the size (diameter) and present a histogram illustrating the size distribution similar to the DLS histogram. Additionally, in Figure 1(c), please ensure that the unit is correctly revised from µM to µm.

In relation to Figure 2, please include a titration curve that graphically represents the Abs (or FL intensities) as the concentration of Au(III) ions varies. Furthermore, please discuss the equivalent and end points for the sensor applications.

The impact of the fluorescence sensor for gold on biological systems remains unclear. Therefore, it is necessary to provide a detailed discussion on the impact of monitoring Au(III) ions in biological systems in both the introduction and discussion sections.

Round 3

Reviewer 1 Report (New Reviewer)

The author present their difficulties in preforming control experiment within a such short time. With revisions regarding the requests from mine and other reviewers, the manscript has been improved. 

It can be accepted to this journal now.

English was improved, good for publication now

Author Response

Reviewer 3 Report (New Reviewer)

Thank you for the corrections and revisions made in the manuscript. However, I am unable to comprehend why the authors state “it is not possible to obtain the results of the size distribution study using SEM”. It should be relatively straightforward to measure the sizes in the previously observed images and create a histogram based on the physical size. I suggest measuring the physical size of particles and discussing the size distribution while comparing it with DLS measurements in terms of hydrodynamic radius.

Regarding the new graph of Abs and FL intensity as a function of [Au(III)] provided in the supporting information, there appears to be a significant linear relationship. I kindly request a discussion on the detection limit and the saturation level of this sensor's performance.

Author Response

This manuscript is a resubmission of an earlier submission. The following is a list of the peer review reports and author responses from that submission.

Round 1

Reviewer 1 Report

This is an interesting paper that presents a new method for determined the Au3+ ions in water using newly synthesized beta-cyclodextrin-encapsulated rhodamine microspheres. Taking into account the toxicity of the above cations, this paper is certainly publishable. The results are well documented. Additional data are presented in the Supplementary information.

There are, however, some points that should be considered by the authors.

1.       The authors named the beta-cyclodextrin-dye microspheres as “core-shell”. However, as far as I understand, the interior and the palisade consist of the same compound.

2.       Some words should be said about the mechanism of the reaction with Au3+ ions. Obviously, the complex formation takes place with participation of three nitrogen atoms accompanied with the rupture of the C–N bond.  What is the role of the cyclodextrin? What part of the dye is included into the host molecule?

3.       In what form the Au3+ ions were introduced in the working solutions?

4.       What apparatus was used for the DLS measurements?

5.       How look the size distributions by volume and particle number?

6.       What equation was used for calculation of the zeta potential?

7.       What is the probable origin of the negative charge of the microspheres?  

8.       What can be said about the repeatability of the synthetic procedure?  

9.       Fig. 1. Is it necessary to demonstrate the negative image (d)?

10.    Last but not least. The text in Lines 139–146 and Figure S8 is unclear. What does the ratio 1 : 1 mean? How can the ligand concentration be expressed? It seems to me that this part of the paper is misleading in the present form.

Reviewer 2 Report

In this work entitled "β-Cyclodextrin-Encapsulated Rhodamine Core–Shell Micro-spheres–Based Fluorescent Sensor for Au3+ and Template for Generating Microplates of Gold" devoted by Munisamy et al., the authors have a fluorescent complexed materials for bioimaging applications. This work deserves publication after major revisions.

1. The optical parameters of the material including molar extinction coefficient and fluorescence quantum yield should be measured.

2. The subcellular localization of the fluorescence should be confirmed via the colocalization experiments with the commercialized probes for various organelles.

3. Control experiments should be provided in which the cells were solely treated with β-CD@RH.

4. In Figure 2, the absorption increased by > 10-fold (Figure 2b) after the addition of Au3+, while the emission only increased by ~4-fold. The authors should check it and explain it.

Reviewer 3 Report

This paper developed β-cyclodextrin-encapsulated rhodamine core–shell microspheres (β- CD@RH) to improve their aqueous solubility and biocompatibility. The β-CD@RHcore–shell microspheres exhibited bright and stable fluorescence with Au3+ ion in aqueous media. This work provides new insight into the significance of designing fluorescent sensor material for monitoring Au3+ ions. Accordingly, I consider the work will be useful in identify the contamination in biological systems, but at present it can not be accepted by this journal, the reason as following:

1. In this paper have shown that “the β-CD@RH core–shell microspheres exhibited excellent selectivity toward Au3+”what the reason for the selectivity, on the other word you need to do some experiments to make sure the way in which materials and Au3+ are combined? As there two parts in theβ-CD@RH core–shell microsphere, which one is the mainly binding site?

2. As the fluorescent sensor material will be used in biological systems, how about the fluorescent sensor material metabolic pathways? Is it safe for the living cells? In the cells there were many different biomolecules,such as DNA, protein, lipid, how does these biomolecules effected the fluorescent sensor to recognize the Au3+?

This paper developed β-cyclodextrin-encapsulated rhodamine core–shell microspheres (β- CD@RH) to improve their aqueous solubility and biocompatibility. The β-CD@RHcore–shell microspheres exhibited bright and stable fluorescence with Au3+ ion in aqueous media. This work provides new insight into the significance of designing fluorescent sensor material for monitoring Au3+ ions. Accordingly, I consider the work will be useful in identify the contamination in biological systems, but at present it can not be accepted by this journal, the reason as following:

1. In this paper have shown that “the β-CD@RH core–shell microspheres exhibited excellent selectivity toward Au3+”what the reason for the selectivity, on the other word you need to do some experiments to make sure the way in which materials and Au3+ are combined? As there two parts in theβ-CD@RH core–shell microsphere, which one is the mainly binding site?

2. As the fluorescent sensor material will be used in biological systems, how about the fluorescent sensor material metabolic pathways? Is it safe for the living cells? In the cells there were many different biomolecules,such as DNA, protein, lipid, how does these biomolecules effected the fluorescent sensor to recognize the Au3+?